# Preliminary characterization of *Plasmodium vivax* sporozoite antigens as pre-erythrocytic vaccine candidates

**Justin Nicholas[1,2], Sai Lata De[1¤a], Pongsakorn Thawornpan[3], Awtum M. Brashear[1,4¤b], Surendra Kumar Kolli[1], Pradeep Annamalai Subramani[1], Samantha J. Barnes[1], Liwang Cui[1,4], Patchanee Chootong[3], Francis Babila Ntumngia[1], John H. Adams[1] ***

**1** Center for Global Health and Interdisciplinary Research, College of Public Health, University of South Florida, Tampa, Florida, United States of America, **2** Department of Molecular Medicine, Morsani College of Medicine, University of South Florida, Tampa, Florida, United States of America, **3** Department of Clinical Microbiology and Applied Technology, Faculty of Medical Technology, Mahidol University, Bangkok, Thailand, **4** Division of Infectious Diseases, Department of Internal Medicine, Morsani College of Medicine, University of South Florida, Tampa, Florida, United States of America

¤a Current address: Department of Infectious Disease & Immunology, College of Veterinary Medicine, University of Florida, Gainesville, Florida, United States of America
¤b Current address: Shriners Children's Genomics Institute, Tampa, Florida, United States of America
* ja2@usf.edu

**Data Availability Statement:** All relevant data are within the paper and its Supporting information files.

## Abstract

*Plasmodium vivax* pre-erythrocytic (PE) vaccine research has lagged far behind efforts to develop *Plasmodium falciparum* vaccines. There is a critical gap in our knowledge of PE antigen targets that can induce functionally inhibitory neutralizing antibody responses. To overcome this gap and guide the selection of potential PE vaccine candidates, we considered key characteristics such as surface exposure, essentiality to infectivity and liver stage development, expression as recombinant proteins, and functional immunogenicity. Selected *P. vivax* sporozoite antigens were surface sporozoite protein 3 (SSP3), sporozoite microneme protein essential for cell traversal (SPECT1), sporozoite surface protein essential for liver-stage development (SPELD), and M2 domain of MAEBL. Sequence analysis revealed little variation occurred in putative B-cell and T-cell epitopes of the PE candidates. Each antigen was tested for expression as refolded recombinant proteins using an established bacterial expression platform and only SPELD failed. The successfully expressed antigens were immunogenic in vaccinated laboratory mice and were positively reactive with serum antibodies of *P. vivax*-exposed residents living in an endemic region in Thailand. Vaccine immune antisera were tested for reactivity to native sporozoite proteins and for their potential vaccine efficacy using an *in vitro* inhibition of liver stage development assay in primary human hepatocytes quantified on day 6 post-infection by high content imaging analysis. The anti-PE sera produced significant inhibition of *P. vivax* sporozoite invasion and liver stage development. This report provides an initial characterization of potential new PE candidates for a future *P. vivax* vaccine.

**Funding:** This study was supported by the National Institutes of Health grants U01AI155361 to JHA and U19AI089672 to LC. U01AI155361 provided partial salary support to JHA, JN, SLD, PT, SKK, PAS, SJB, PC, and FBN. U19AI089672 provided partial support to LC and AMB. The funders had no role in study design, data collection and analysis, decision to publish, or preparation of the manuscript.

**Competing interests:** The authors have declared that no competing interests exist.

## Author summary

*Plasmodium vivax* infections are the most common cause of malaria outside of Sub-Saharan Africa and often prevalent in areas with limited healthcare. A vaccine that interrupts parasite transmission from the mosquito would help limit the burden of vivax malaria. The pre-erythrocytic stages represent an important bottleneck in the life cycle prior to the disease-causing blood stage. However, there is limited knowledge of sporozoite and liver stage antigens that might be suitable vaccine candidates. To address this gap, we identified and characterized sporozoite antigens that could potentially be targets of a pre-erythrocytic stage vaccine. We successfully evaluated three novel candidates as targets of inhibitory antibody responses against liver stage infection and development. Data from this study supports advancement of these targets for further *P. vivax* vaccine development.

## Introduction

Malaria is a major public health concern with approximately 247 million cases and 619,000 deaths occurring in 2021 in 84 countries worldwide [1]. *P. vivax* infections account for a large portion of reported malaria cases in the Americas (71.5%), South-East Asia (39.7%), Western Pacific region (31.9%), and the Mediterranean Region (18%) [1]. Clinically, *P. vivax* has been mischaracterized as a benign disease, since infections often cause intense tertian malaria fever or paroxysm, anemia, repeated relapse infections, and low birthweight [2,3]. Moreover, some liver stages can pause development to become dormant hypnozoite stages that later can activate to complete development to blood stages leading to clinical relapse infections furthering disease morbidity and transmission.

Development of an effective pre-erythrocytic (PE) stage vaccine that targets the infecting sporozoites from the mosquito and development of liver stages offers the potential to neutralize a new infection of *P. vivax* sporozoites before any clinical illness or subsequent transmission to mosquitoes. In particular, the non-replicating short-lived sporozoites are directly exposed to antibodies during their migration to the liver while the resident liver stages have prolonged vulnerability to immune effector cells. Therefore, a PE vaccine can possibly prevent clinical disease, hypnozoite formation, and transmission [4,5]. However, functional protective responses of *P. vivax* PE antigens are poorly characterized relative to *P. falciparum* [6]. Currently, this is a major gap in our understanding of potential *P. vivax* vaccine candidates.

The *Plasmodium* circumsporozoite protein (CSP) is the dominant sporozoite surface protein and the long-time leading PE vaccine candidate based on early studies that linked immune responses to CSP to the induction of sterile immunity by irradiated *P. berghei* sporozoites [7–10]. RTS,S/AS01 and R21/MM are virus-like nanoparticle vaccines containing components of the *P. falciparum* circumsporozoite protein (PfCSP) [11–13]. When used seasonally in combination with chemoprevention, RTS,S reported efficacy is 70.6% against severe malaria while R21 is 81% effective against clinical malaria in children [13,14]. However, similar progress toward a *P. vivax* CSP-based vaccine were not successful in initial clinical trials [15–22]. Additional studies suggest a lack of long-lived protective responses of PvCSP may be due to poor immunogenicity [23,24]. Thus, there is a need to identify other potential PE candidates for inclusion with PvCSP in a multivalent vaccine design to prevent infection and subsequent progression to clinical disease and mosquito transmission.

Ideal antibody-based PE vaccine candidates would be surface-exposed sporozoite antigens accessible to immune antibodies that could act synergistically with PvCSP or other antigens during the infection process. The ideal PE vaccine would inhibit gliding as the sporozoite

migrates from the skin to the liver, block sporozoite cell traversal when it reaches the liver parenchyma, inhibit invasion of and development in hepatocytes, and block egress to initiate blood stage development. Previously, we identified numerous *P. vivax* PE antigens that were upregulated in response to mammalian host-like microenvironmental changes that enhanced sporozoite infectivity [25]. These potential PE candidates that associated with infectivity were analyzed for other favorable properties, such as other evidence for essentiality to invasion and/or liver stage development to select to a short list of four antigens to evaluate as potential PE vaccine candidates.

Sporozoite surface protein 3 (SSP3) was first characterized by Harupa et al in *P. yoelii*. As gliding plays an important role in sporozoite infectivity [26] it was found that PySSP3⁻ mutant sporozoites were gliding deficient [27]. In a related study, the PbSSP3⁻ mutant sporozoites had normal gliding patterns, yet mutants were unable to cause a blood-stage (BS) infection [28]. This discrepancy demonstrates that mutant SSP3 orthologs in rodent species of *Plasmodium* have varying phenotypes. A second vital PE antigen is the sporozoite surface protein essential for liver stage development (SPELD). In *P. berghei*, PbSPELD⁻ mutants resulted in early PE developmental arrest, yet other invasive phenotypes such as gliding were unaffected [29]. Another essential sporozoite microneme protein, MAEBL, is vital for sporozoite salivary gland invasion, and hepatocyte invasion [30–33]. Antisera targeting the M2 domain of MAEBL demonstrated protection against inhibition of PE development and lethal *P. yoelii* infection [30,34]. Lastly, we selected the sporozoite protein essential for cell traversal (SPECT1), which is another microneme protein that facilitates sporozoite transmigration across host cells [35–37]. These antigens were chosen for likelihood of antibody accessibility, their important roles in sporozoite invasion, and subsequent liver stage development. Here, we partially characterize these novel *P. vivax* PE antigens providing preliminary results to support their potential as *P. vivax* PE vaccine targets.

## Materials and methods

### Ethics statement

Protocols for human subjects were approved by the Ethics Committee on Human Rights Related to Human Experimentation, Mahidol University Thailand [MU-IRB 2012/079.2408]. Written informed consent was obtained from all participants. BALB/c mice aged 4–6 weeks were maintained under pathogen-free conditions per Institutional Animal Care and Use Committee (IACUC) protocol R IS00007010 approved by the University of South Florida Ethics Committee.

### Antigen production, expression, and purification of antigens

*Plasmodium vivax* gene sequences coding for SSP3 (PVX_123155), SPECT1 (PVP01_1212300), SPELD (PVX_092505), and M2-MAEBL (PVP01_0948400) were acquired from PlasmoDB [38]. The signal peptide of PvSSP3 and PvSPECT1 as well as the transmembrane domain for PvSSP3 and PvSPELD were excluded in expressed recombinant proteins (Fig 1b). Modifications were done to ensure that the functional domains remained intact while hydrophobic and disordered regions were removed (Fig 1b and S1 Dataset). Relative molecular weight and predicted isoelectric points were estimated using ExPASy (Compute pI/Mw tool). The modified coding sequences were codon-optimized for *E. coli* expression (Genscript, USA) and cloned into a pET21a+ expression vector with a C-terminal 6xHis-tag. Recombinant antigens were expressed in One Shot BL21 Star (DE3) *Escherichia coli*, henceforth referred to as BL21 (Invitrogen).

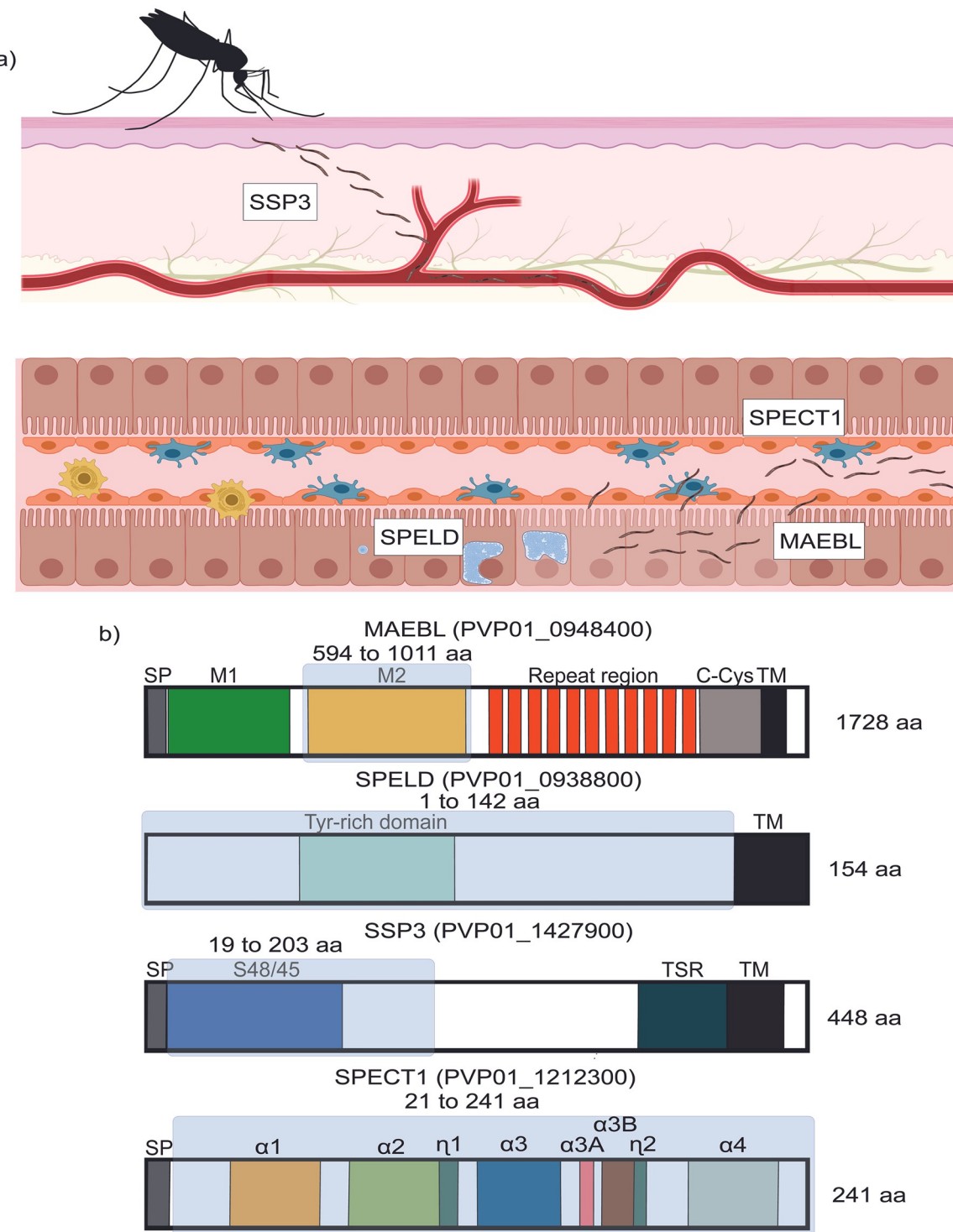

**Fig 1. Critical functions of sporozoite invasion facilitated by antigens of interest.** a) Schematic of skin-to-hepatocyte stage demonstrating the functional role of antigens in invasive sporozoites. Created with BioRender.com b) *P. vivax* PE antigens chosen for overexpression. Schematic of protein constructs (not drawn to scale) with putative domains and secondary structures identified. Highlighted sections in blue boxes represent the estimated domain/region chosen for recombinant expression in BL21.

Starter cultures (350ml) of *E. coli* expressing clones were grown overnight. The next day, the starter culture was diluted to 20% in 4.2 L of warm LB broth. The bacterial broth was incubated at 37˚C, 250 rpm until an OD value of 0.6 to 0.7 and induced with IPTG (Goldbio) to a final concentration of 1 mM. The induced culture was left to incubate at 30˚C, 250 rpm for 6–8 hours, and the bacterial pellet was harvested by centrifuging at 10,000 rpm.

To determine the solubility of the expressed protein, the bacterial pellet was lysed with phosphate buffer (0.5M NaCl, 20 mM sodium phosphate buffer, and 25 mM Imidazole) and purified by affinity chromatography using the HisTrap HP columns (GE) on the Akta Pure purification system (Cytiva). Western blots were performed with an anti-His tag antibody to confirm expression of each recombinant protein. The insoluble fractions were purified from inclusion bodies (IB). Briefly, the remaining bacterial pellet was lysed using two separate buffers, buffer 1 (50 mM Tris, 0.5M NaCl, 0.2 mM EDTA, 3% sucrose, and 1% Triton X-100) and buffer 2 (50 mM Tris, 0.5M NaCl, 0.2 mM EDTA and 3M urea) and IB was recovered by centrifugation. The IB was solubilized in a 20mM phosphate buffer containing 6M guanidine hydrochloride and recombinant protein purified by affinity chromatography and refolded by rapid dilution. SDS-PAGE was used to analyze the native formation of refolded protein by observing protein migration in reducing and non-reducing conditions. SeeBlue Plus2 Pre-Stained Protein Standard (ThermoFisher) and Precision Plus Protein Dual Color Standard (BioRad) were used to evaluate relative molecular weight. Endotoxins were removed from proteins using a ToxinEraser Endotoxin Removal kit (GenScript).

## Variant calling and epitope analyses

High-quality variant calls were extracted from a previously published dataset of Southeast Asian *P. vivax* samples as described [39] (S2 Dataset). FASTA sequences for all antigens were sourced from NCBI via a MegaBLAST restricted to *P. vivax* (taxid:5855). To expand our dataset, we searched available genome assemblies for antigen sequences. 4 published draft genomes had corresponding genes extracted and 19 genome assemblies which had been created as described in [40] up through correction were searched via blast for similarity to PVP01_0938800, PVP01_0948400, PVP01_1212300, and PVP01_1427900. Other published genomes including PvC01, PvT01, PvW1, PvCMB-1, and Pv_N_Korean_v1 were obtained from respective repositories and similarly BLAST searched [41–43]. Sequences were aligned in MEGA 11 [44] and formatted to remove gaps and ambiguous sequences. Site-specific nonsynonymous and synonymous substitution rates (*dN/dS*) were analyzed for selective pressure on Datamonkey webserver (Datamonkey Adaptive Evolution Server) (S3 Dataset).

Epitopes were predicted using Immune Epitope Database (IEDB) bioinformatic tools [45]. Linear B-cell epitopes were predicted using BepiPred 2.0 [46]. MHC-I binding predictions were done using SMMPMBEC methodology on supertype alleles HLA-A*01:01, HLA-A*02:01, HLA-A*03:01, HLA-A*11:01, HLA-A*24:02, HLA-B*7:02 [47]. MHC-II supertypes were taken from Table 2 of Greenbaum et al. and predicted using SMM-align (NetMHCII 1.1) method [48] for all antigens. T cell epitopes that had an $IC_{50}$ lower than 50 nM were selected as they are predicted to be high affinity binders (S4 Dataset).

## Mouse immunization

Groups of 5–10 inbred (BALB/c) mice were immunized subcutaneously on days 0 and 21 with 25 µg of each antigen emulsified in TiterMax Gold (1:1) (TiterMax USA, Inc). Control groups received an emulsification of TiterMax Gold and PBS (1:1). On day 42, a final 25 µg of antigen dose in PBS was administered subcutaneously. Pre- immune (Day 0) and immune sera (days 14, 35, and 56) were collected to quantify the antibody titers.

## Enzyme-Linked immunosorbent assay (ELISA)

Nunc 96-well flat-bottom microtiter plates were coated with 2 μg/ml of the desired antigen in carbonate coating buffer (0.0125 M $NaHCO_3$, 0.0875 M $Na_2CO_3$ at pH 9.6) overnight at 4˚C. Plate wells were blocked with 5% non-fat milk in PBS with 0.05% Tween (PBST) for 2 hours at room temperature (RT). Plates wells were then incubated in duplicate for 2 hours at RT with serial dilution of mouse sera (starting at 1:100). Bound immune sera were detected using Blue-Phos Microwell Phosphatase Substrate System (KPL) measured at 650 nm absorbance.

To determine total IgG antibody levels in patient plasma, the plasma was tested at 1:200 dilutions. The protocol used was similar to that used for the mouse sera except goat anti-human IgG-peroxidase conjugated secondary antibody (Sigma) was used for detection. A baseline OD was established using plasma samples from the 30 healthy control (HC) volunteers. Total IgG antibody levels were reported as reactivity index (RI), calculated by dividing OD values of tested samples by a cut-off value (mean + 2 s.d.) from HC samples. An RI $\geq$ 1 was considered responders while RI $<$ 1 were considered non-responders.

Cryopreserved *P. vivax* sporozoites were thawed at RT, sonicated, and concentration determined by Nanodrop. The lysate was used to coat Nunc 96-well flat-bottom plates at 10 μg/ml in bicarbonate coating buffer overnight at 4˚C. Monoclonal antibodies (mAbs) purified from hybridoma cell lines obtained from BEI resources, NIAID, NIH: (MRA-183) 2A10 Anti-*Pf*CSP, (MRA-184), 2F2 Anti-*Pv*CSP VK210, (MRA-185) 2E10.E9 Anti-*Pv*CSP VK247, and (MRA-100) 3D11 Anti-*Pb*CSP were used to determine sporozoite species at 1 μg/ml (S1 Fig). Sporozoite lysate was also tested for reactivity with immune mouse sera raised against the various antigens by ELISA as described for the recombinant antigens above.

## Immunofluorescence assay (IFA)

Cryopreserved *P. vivax* sporozoites were thawed at RT, and quantified using a hemocytometer. Sporozoites were spotted on a specimen slide (Tekdon) at a density of $1 \times 10^4$ sporozoites per well. The slides were air dried and fixed with 4% paraformaldehyde (PFA) for 15 minutes and washed thrice with PBS. The fixed-sporozoites were blocked in 5% fetal bovine serum (FBS) in PBS for 45 minutes at RT and then incubated overnight at 4˚C with mouse polyclonal antibodies (pAbs) of rPvSPECT1, rPvSSP3 or rPvM2-MAEBL at 1:500 dilution. The slides were washed thrice and incubated with Alexa Fluor 488 goat anti-mouse IgG secondary antibody (1 μg/ml, Invitrogen) and Hoechst 33342 (1:1000 dilution, Fisher Scientific) for 1 hour at RT. ProLong Glass Antifade Mountant (Thermo Scientific) was used to preserve signal intensity during image acquisition. pAbs to Duffy binding protein (DBPII) served as an isotype control and anti-PvCSP VK210 (2F2) mAbs served as a positive control. Sporozoites were examined under oil immersion at x100 magnification using the Zeiss AXIO Observer.Z1 (ZEISS Group).

## Study participants

Regions of low malaria transmission along Southern Thailand in the Ranong and Chumphon provinces were chosen to recruit *P. vivax*-infected volunteers (S1 Table). The volunteers (n = 52) were diagnosed by microscopic examination of both thin and thick blood films and confirmed by nested PCR. Plasma samples were collected to test the reactivity of total IgG to recombinant antigens. Volunteers were a mixed group of Thai (n = 41) and Myanmar (n = 11) nationals, ages ranging from 18 to 63 years old (S1 Table). Naïve control plasma (n = 30) was obtained from volunteers in a non-malaria endemic region (Bangkok, Thailand).

## Inhibition of liver stage development assay (ILSDA) with cryopreserved *P. vivax* sporozoites

Mouse sera inhibition of *P. vivax* sporozoite invasion and development was assessed using primary human hepatocytes (PHH) donor lot YNS (Bioreclamation IVT). Cryopreserved *P. vivax* sporozoites were thawed as previously described in hepatocyte culture medium (HCM) (InVitroGro CP Medium) at RT [49]. $2x10^4$ sporozoites were incubated with pooled and heat inactivated mouse sera at 1:50 dilution in HCM for 20 minutes prior to adding to wells of PHH seeded plate in triplicates. Infected plates were centrifuged at 200 x g for 5 minutes and sporozoites allowed to invade at 37˚C for 16 hours before changing the medium. PHH medium was changed every alternate day until day 6 post infection and the cells were fixed with 4% PFA. Non-treated sporozoites served as infection control for normalization while anti-Duffy Binding protein polyclonal mouse sera served as a non-specific control.

Fixed *P. vivax* schizonts were incubated overnight with anti-UIS4 polyclonal rabbit sera in 0.22 micron filtered dilution buffer (1% (w/v) BSA, 0.3%, Triton X-100 in PBS) at 1:5000 as previously described [50]. After overnight incubation plates were washed twice with PBS and incubated with Alexa Fluor 488 goat anti-rabbit IgG secondary antibody (1 μg/ml, Invitrogen) and Hoechst 33342 (1:1000 dilution, Fisher Scientific) for 1 hour at RT. The plates were washed with PBS and imaged at x20 magnification.

Liver-stage (LS) forms high content imaging were accomplished using the Cell Insight CX7 system and the HCS studio software via a modified Target Activation Advanced Bioapplication (Thermo Fisher Scientific). LS forms were counted using the Alex Fluor 488 channel and identified by mean intensity and cell roundness as previously described [50]. Wells with no test sera exposed sporozoites served as controls for % inhibition calculations.

$$\% \, Inhibition = 100 - \left[ \frac{x}{y} \times 100 \right]$$

Where '$x$' is average number of LS forms in wells with sera exposed sporozoites and '$y$' is average number of LS forms in control wells previously described [50].

## Statistical analyses

ELISA data were analyzed using GraphPad Prism v9, GraphPad (San Diego, USA). A baseline OD was established using plasma samples from the 30 healthy controls (HC). Total IgG for each antigen was standardized to a reactivity index (RI), calculated by dividing the OD values of tested samples by a cut-off value (mean + 2SD) from HC samples. An RI $\geq$ 1 was positive for specific antibodies; those with an RI $<$ 1 were considered negative. RI was tested for normality using the Shapiro-Wilk test. A Kruskal-Wallis one-way ANOVA with a Dunn's multiple comparisons test was used to compare seropositivity between groups.

Mouse endpoint titers were tested for normality using D'Agostino & Pearson test. Analysis of mouse endpoint titers to sporozoites, antisera inhibition, and LS form area were done using Kruskal-Wallis one-way ANOVA with Dunn's multiple comparisons test as the groups were not normally distributed. For selection pressure analyses, we used the fixed effects likelihood (FEL) [51] and mixed effects model of evolution (MEME) [52] with a threshold p-value of p$\leq$ 0.05 after removing identical sequences.

## Results

### Expression of *P. vivax* PE antigens in bacteria

The selected *P. vivax* sporozoite antigens represent the migratory-to-hepatocyte-infecting phases of the PE infection processes (Fig 1a and Table 1). Signal peptides and other potentially hydrophobic regions were not included as part of the recombinant proteins. Coding sequences of the PE candidates were expressed in bacteria as near full-length recombinant proteins (rPvSPELD, rPvSPECT1) or the primary functional domains of the PE candidates (rPvMAEBL, rPvSSP3) (Fig 1b). In particular, the expressed fragments of rPvSSP3 included the predicted S48/45 functional domain, and rPvMAEBL contained the M2 PAN-like putative ligand domain. The full-length ectodomain of rPvSPECT1 retained all four helix bundles and the ectodomain of rPvSPELD contained its tyrosine-rich region (Fig 1b). Further, we tested feasibility for future product development as a manufactured recombinant protein product using the established protocol developed for production of a conformationally-correct cysteine-rich ligand domain of the *P. vivax* Duffy binding protein (DBP), also known as DBPII, a blood-stage vaccine candidate [53]. In bacterial expression rPvSPECT1 could be directly purified from the soluble fraction of clarified bacterial lysate, while rPvSSP3, rPvSPELD and rPvM2-MAEBL were purified from inclusion bodies, and refolded to native conformation by rapid dilution. The purified recombinant proteins of rPvSSP3, rPvSPELD and rPvSPECT1 expressed at the expected sizes of 23.0 kDa, 20.3 kDa and 25.7 kDa compared to standard protein ladder when analyzed by SDS-PAGE (Fig 2a–2c) with average yields of 12 mg/L, 7.25 mg/L, and 2.85 mg/L, respectively. Some rPvSPECT1 was present in inclusion bodies, but most of the expressed recombinant antigen was in the soluble fraction. rPvM2-MAEBL migrated at ~64 kDa, over 30% higher than its predicted 48.3 kDa (Fig 2d) and had an average yield of 1.9 mg/L. Unlike rPvM2-MAEBL and rPvSSP3, rPvSPELD aggregates after purification and refolding from inclusion bodies, leading to its exclusion from further characterization.

### Immunogenicity studies generated high-titer antibodies that recognized antigens on cryopreserved *P. vivax* sporozoites

To determine whether the immune sera raised against the recombinant proteins recognized *P. vivax* sporozoites, we performed an ELISA with crude sporozoite lysate as well as indirect immunofluorescence with sporozoites. The murine antisera to the recombinant PE antigens recognized epitopes on *P. vivax* sporozoites (Fig 3). Antibody endpoint titers were determined (sera from day 56) by reactivity to whole sporozoite lysate (Fig 3a). Titers ranged from 1:200 to 1:6400 and PvSPECT1 had significantly higher reactivity to sporozoites than PvM2-MAEBL

**Table 1. Key attributes of *P. vivax* PE antigens to prioritize as potential vaccine candidates.**

| Attribute | SPECT1 (PVP01_1212300) | SSP3 (PVP01_1427900) | M2-MAEBL (PVP01_0948400) | SPELD (PVP01_0938800) |
|---|---|---|---|---|
| **Expression in the PE stage** | All antigens are highly expressed in the PE stage of *P. vivax* [25,54,55]. | | | |
| **Antigen localization** | Sporozoite microneme [35] | Salivary gland sporozoite surface [27,28] | Sporozoite microneme/ surface and BS merozoite rhoptry [30,32,56,57] | Midgut sporozoite surface to early PE development [29] |
| **Putative function** | Cell traversal [35–37] | Gliding, cell traversal (Py) [27], and PE development (Pb) [28] | Salivary gland and host hepatocyte invasion [30,34] | PE development [29] |
| **Evidence supporting immune targeting** | Antisera to PfSPECT1 found in RAS-protected individuals [58] | - | Functional antibodies confer protection in PE and BS [30,34] | PbSPELD *ko* sporozoites arrest in mid liver stage induced 50% efficacy [29] |

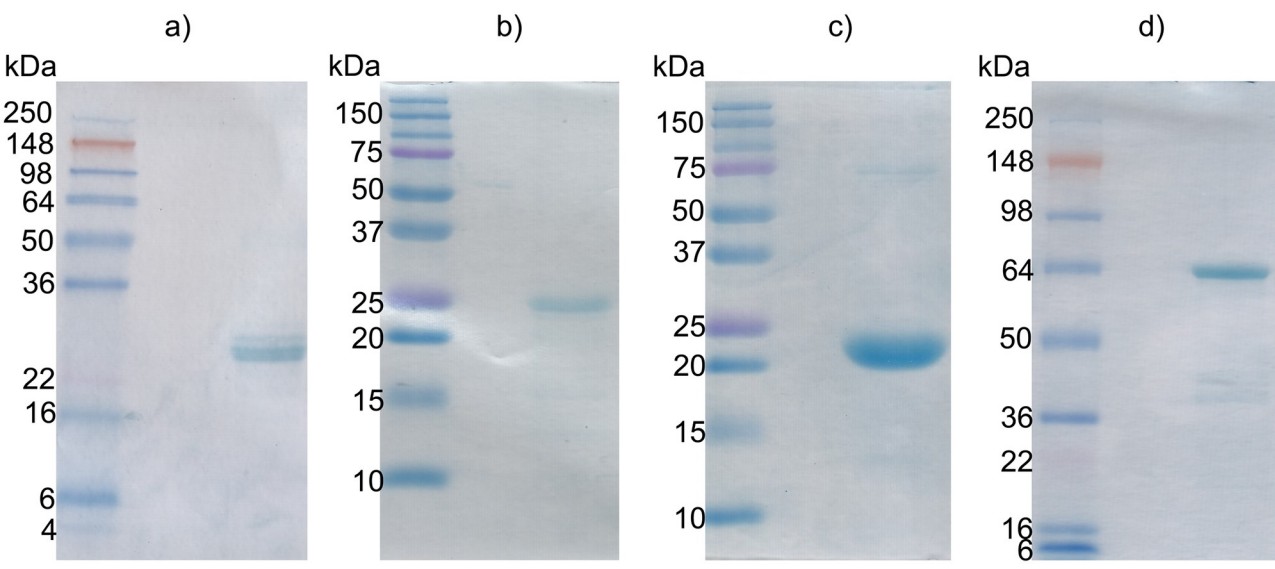

**Fig 2. 12.5% to 15% SDS-PAGE gels of purified recombinant proteins expressed in *E. coli*.** Purified and refolded IB antigens of a) ~1 µg rPvSSP3. b) ~1 µg rPvSPELD c) ~2 µg of soluble rPvSPECT1 d) ~1 µg rPvM2-MAEBL.

(p = 0.005) (Fig 3a), which might reflect relative abundance or epitope accessibility in the crude antigen extract instead differences in titer. Indirect immunofluorescent revealed the expected surface-association with PvSSP3 and PvSPECT1 (Fig 3b and 3c) while the more intense anterior end fluorescence observed for PvM2-MAEBL was consistent with localization to the micronemes (Fig 3d).

## Mouse antisera inhibit invasion and attenuate growth of *P. vivax* sporozoites in PHH

The potential vaccine efficacy of the PE antigens was evaluated using an *in vitro* assay for inhibition of liver stage development (ILSDA). The 384-well liver assay uses primary human hepatocytes to support high level of infections with *P. vivax* and *P. falciparum* sporozoites leading to complete liver development to blood stage breakthrough [50]. Each antiserum to the recombinant PE antigens demonstrated significant inhibitory activity against *P. vivax* sporozoite invasion into hepatocytes as quantified by high content imaging at day 6 post-infection relative to control (Fig 4a). Antisera to PvSPECT1, PvSSP3 and PvM2-MAEBL inhibited *P. vivax* sporozoite invasion by 31%, 30%, and 25% respectively (Fig 4a).

In addition, a decreased growth phenotype was observed in developing LS forms relative to the no serum control for all antisera tested (Fig 4b). Surprisingly, sporozoite pre-incubation with the anti-DBPII serum also resulted in growth attenuation of the *P. vivax* liver stages, which might reflect some toxicity in the mouse serum or cross-reactivity with a sporozoite antigen. Additionally, the proportion of developing LS small forms ($<100$ µm$^2$) to large forms ($>100$ µm$^2$) were similar between groups in the donor lot tested for all experimental groups (S3 Fig).

## Post-infection serological responses to PE candidate antigens

We next evaluated whether residents in a malaria endemic region in Thailand with natural exposure to *P. vivax* had serological reactivity against the recombinant PE candidate antigens versus healthy controls (HC). Of 52 samples analyzed by ELISA, seropositivity was observed in

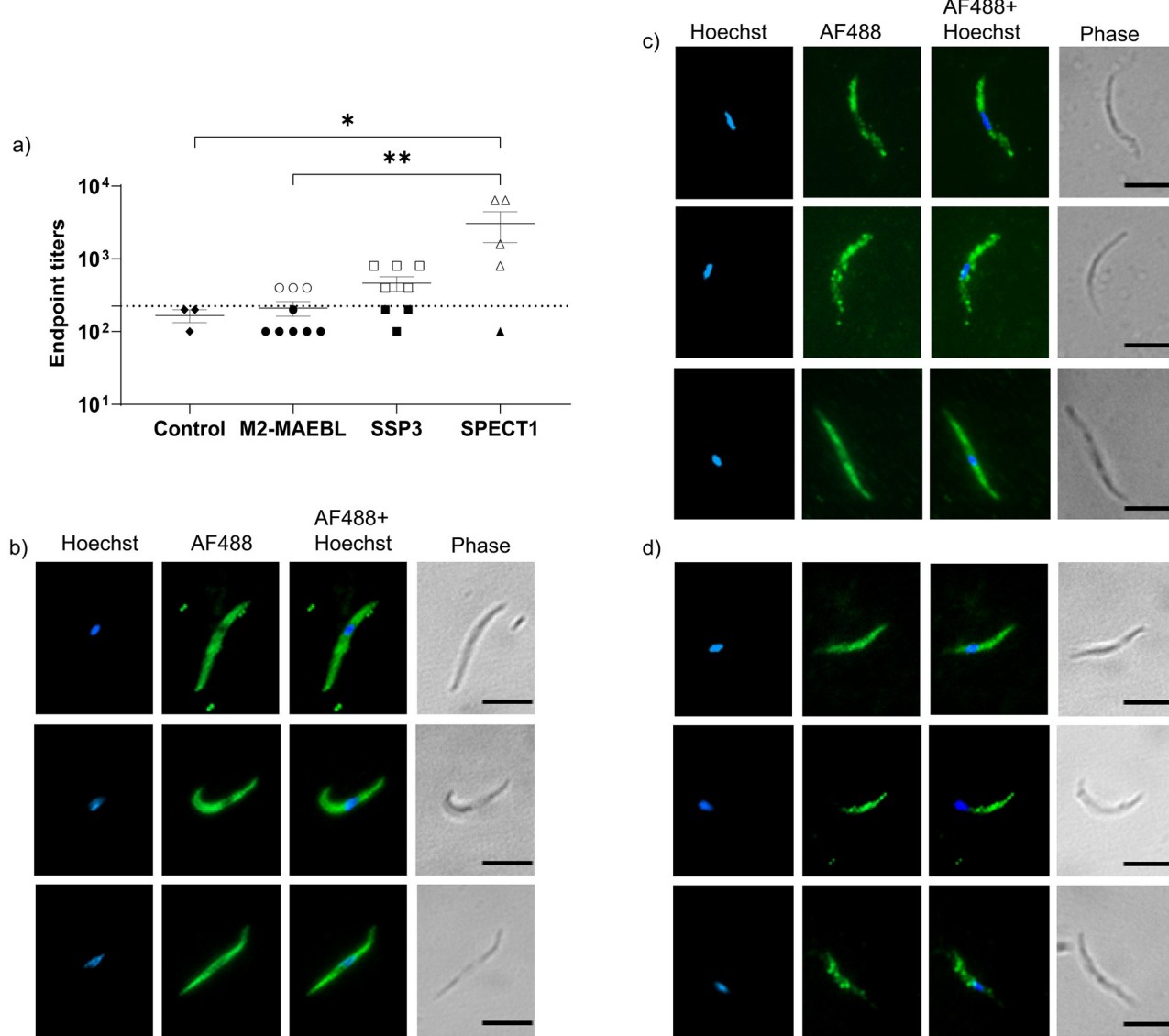

**Fig 3. Antisera reactivity to *P. vivax* sporozoites.** a) Antisera from mice immunized with recombinant SSP3, SPECT1, and M2-MAEBL were reactive to *P. vivax* sporozoite lysate. Dotted line indicates cutoff with values >1 s.d. above the mean of control reactivity considered positive (clear). For indirect immunofluorescence mouse antisera to b) SSP3, c) SPECT1, and d) M2-MAEBL were used as primary antibodies followed by Alexa-Fluor 488 (AF488) goat-anti mouse secondary and Hoechst 33342 (Hoechst). Scale bar (black) indicates 5μm. Statistical significance was determined using a non-parametric Kruskal-Wallis one-way ANOVA with a mean rank multiple comparisons test and represented as p<0.05 (*), p = 0.005 (**).

11 (21.1%), 15 (28.8%), and 17 (32.7%) against rPvM2-MAEBL, rPvSSP3, and rPvSPECT1, respectively (Fig 5). However, significant differences were not observed between antigen responses (Fig 5). Interestingly, only 7.6% of the samples were seropositive for all three antigens and both rPvSSP3 and rPvSPECT1, while 3.8% were reactive to both rPvM2-MAEBL and rPvSSP3 and 1.9% reactive to both rPvM2-MAEBL and rPvSPECT1 (Fig 5).

## Variant calling of genes identifies polymorphisms in putative epitopes

We next identified the polymorphisms by variant calling among sequence information available from the published sequences to characterize residues in regions that may be under

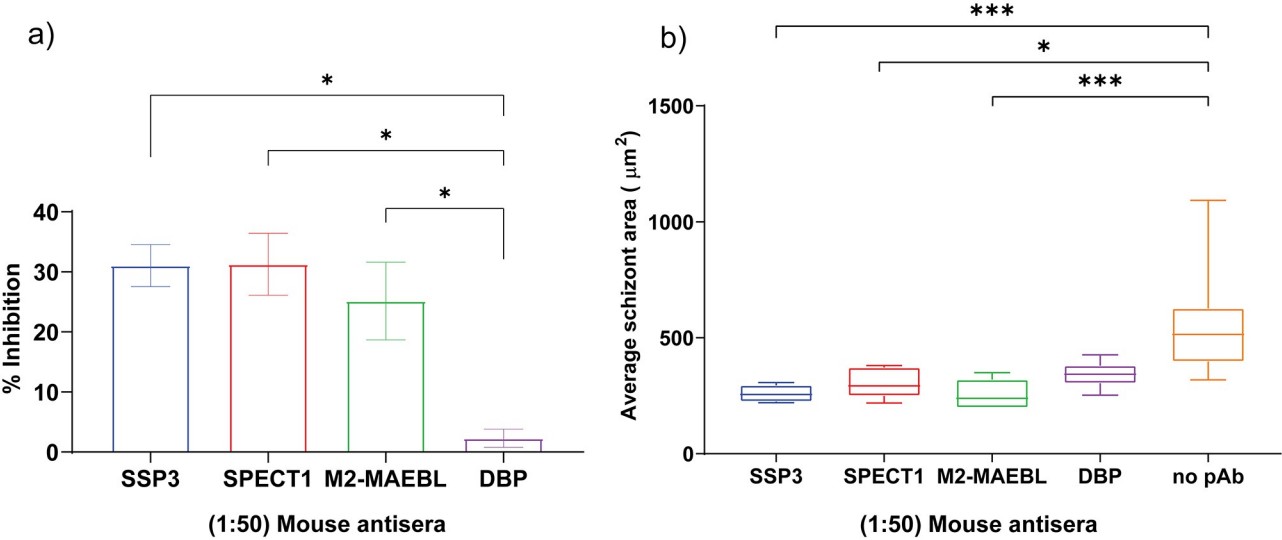

**Fig 4. Inhibition of *P. vivax* liver stage development.** a) Mouse antisera to each PE vaccine candidate significantly inhibited *P. vivax* sporozoite invasion in primary human hepatocytes versus no antibody control and antiserum to *P. vivax* DBP. b) *P. vivax* sporozoites incubated with anti-PE sera significantly decreased in size post-invasion. Small LS forms ($<100 \, \mu m^2$) were removed from all populations prior to analysis. Results were determined on day 6 post-infection by high content imaging of infected hepatocytes by antibody staining of intracellular parasites with UIS4 and are expressed as mean with error bars indicating the standard error of the mean. Statistical significance was determined using a non-parametric Kruskal-Wallis one-way ANOVA with a mean rank multiple comparisons test and represented as $p<0.05$ (*), $p = 0.0002$ (***), and $p<0.0001$ (****).

immune selection. There were only 5 polymorphisms in the M2 domain apart from the 19 other SNPs in the PvMAEBL CDS. Likewise, we observed 4 (PvSSP3) and 1 (PvSPELD) mutation (S2 Dataset). Surprisingly, no mutations were identified within the CDS of PvSPECT1. Therefore, based upon frequency of synonymous to nonsynonymous mutations ($p \leq 0.05$) this analysis suggests on this small sample size that none of the genes for these PE candidates are under strong immune selection (Table 2 and S3 Dataset). Though the observed variation was relatively limited compared to many merozoite surface antigens, further analysis indicated some of these variant residues were observed in putative B cell epitopes in the expressed portions of the PE candidates. In PvSSP3, PvM2-MAEBL, and PvSPELD, we identified 8 (PvSSP3 & PvSPECT1), 12 (PvM2-MAEBL), and 3 (PvSPELD) (S4 Dataset). There were 2 and 114 (PvSSP3), 12 and 140 (PvSPELD), 12 and 146 (PvSPECT1), 13 and 106 (PvM2-MAEBL) putative T cell epitopes for MHC Class I and II, respectively (S4 Dataset). A single point mutation was observed in SPELD (I73T), PvM2- MAEBL (I981S) for the Class II and I epitopes, respectively (Table 2). PvSSP3 had a single point mutation Q174E in a putative B cell epitope, with an uncharged glutamine replaced by a charged glutamic acid (Table 2).

## Discussion

The skin-to-hepatocyte stage transition (Fig 1a) represents a critical bottleneck in the life cycle of *Plasmodium spp.*, since sporozoites are non-replicating and relatively few are injected by a mosquito bite [4,5]. Nonetheless, completion of development in the liver by a single sporozoite can result in a clinical malaria and potentially death. The poor track record in bringing to market effective anti-sporozoite vaccines that can block infection clearly demonstrates the challenge inducing an immune response capable of neutralizing sporozoites. Multiple immune interventions need to be considered to exploit this bottleneck for a designing an effective PE vaccine candidate. Therefore, our goal is to prioritize PE targets critical for sporozoite invasion

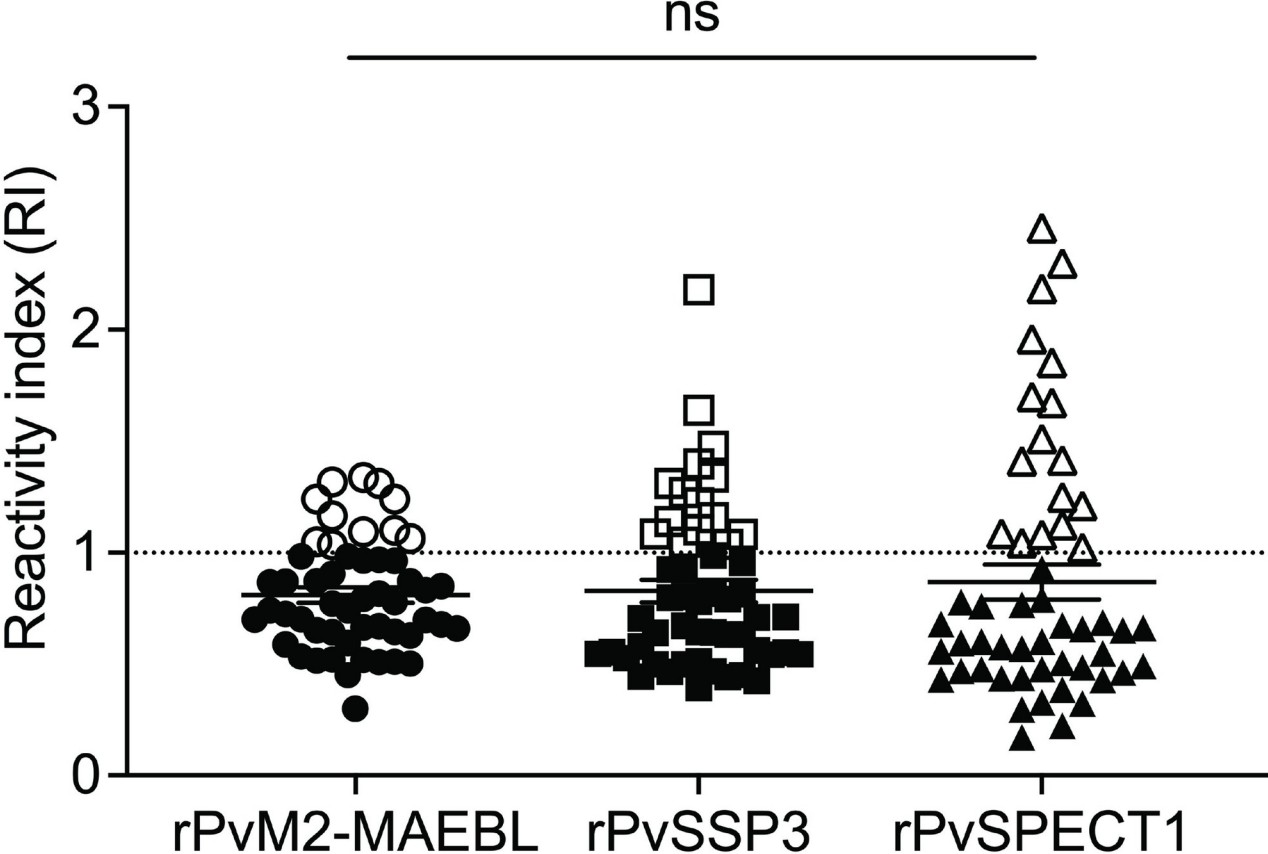

**Fig 5. Seropositivity of plasma samples from *P. vivax*-infected patients (n = 52) in Ranong and Chumphon provinces of Thailand.** The reactivity index (RI) of total IgG responses to rPvM2-MAEBL, rPvSSP3, and rPvSPECT1. Dotted line indicates the cutoff for RI samples $\geq$ 1. Error bars indicate the s.e.m. Statistical significance between responses was tested using a Kruskal-Wallis one-way ANOVA with a mean rank multiple comparisons test and represented as not significant (ns).

that induce a broadly neutralizing, strain-transcending, functionally inhibitory antibody response. Previously, we identified *P. vivax* PE antigens that were upregulated in an infective microenvironment [25]. We further evaluated key characteristics of vaccine candidates with the objective of down-selecting potential PE targets suitable to be incorporated into multivalent antibody-based vaccine effective against *P. vivax* (Table 1). Based on these criteria, PvSSP3, PvM2-MAEBL, PvSPECT1, and PvSPELD were chosen for further characterization in our study.

Here, we successfully expressed recombinant PvM2-MAEBL, PvSSP3 and PvSPECT1 in an established recombinant protein expression system that allows for high-quality cost-effective protein production and is suitable for clinical studies [3,59,60]. Bacterial expression also prevents unwanted post-translational modifications and is relatively easy to manipulate [61]. Previous studies demonstrated that apart from transmembrane and signal peptide regions, greater protein disorder, and a more basic isoelectric point contribute to reduced soluble expression in *E. coli* [62]. Therefore, recombinant antigens were designed by eliminating hydrophobic regions without truncating putative and confirmed domains for PE protein function (Fig 1b).

Despite optimizations to facilitate antigen expression, rPvSSP3 and rPvSPELD had to be purified from inclusion bodies (Fig 2a and 2b) and refolded to achieve 'native' conformation. The rPvSPECT1 expressed as a soluble protein similar to the expression of PbSPECT1 [63].

**Table 2. Polymorphic residues of PvSSP3, PvSPELD, and PvM2-MAEBL in putative epitopes.** Variants in epitopes were determined after a MUSCLE alignment using MEGA. Codon-by-codon selective pressure was statistically tested by FEL and MEME ($p \leq 0.05$) in putative linear B-cell and MHC I/II T-cell epitopes.

| Residue | Peptide | Nucleotide position | Codon | FEL p-value | MEME p-value | Ref (PVP01) | Mutant | AA substitution | HLA Allele | Epitope |
|---|---|---|---|---|---|---|---|---|---|---|
| **PvSSP3** (n = 31) | | | | | | | | | | |
| 165 to 185 | VDLYNQLDLSKDNSNNK | 520 | 174 | 0.18 | 0.36 | CAA | GAA | Q174E | - | B |
| **PvSPELD** (n = 27) | | | | | | | | | | |
| 70 to 84 | ESPIVCLSSKKVIKD | 392 | 130 | 1.00 | 1.00 | ATC | ACC | I73T | DRB1*01:01 | T |
| 63 to 77 | YVYYTPAESPIVCLS | 392 | 130 | 1.00 | 1.00 | ATC | ACC | I73T | DRB1*09:01 | T |
| 70 to 84 | ESPIVCLSSKKVIKD | 392 | 130 | 1.00 | 1.00 | ATC | ACC | I73T | DRB1*11:01 | T |
| **PvM2-MAEBL** (n = 39) | | | | | | | | | | |
| 697 to 712 | MGKRNGRSIELPYDKS | 2099 | 700 | 0.36 | 0.33 | AGA | AAA | R700K | - | B |
| 829 to 840 | IEDTYKNKCFRN | 2491 | 831 | 0.47 | 0.48 | GAC | AAC | D831N | - | B |
| 862 to 887 | RIDNCRKEKTDLSKPNCQKLRKTSDS | 2588 | 863 | 0.30 | 0.34 | ATC | ACC | I863T | - | B |
| 969 to 983 | IANESVNKDNMFIVN | 2942 | 981 | 0.79 | 0.24 | ATT | AGT | I981S | - | B |
| 980 to 989 | FIVNGECYYV | 2942 | 981 | 0.7921 | 0.24 | ATT | AGT | I981S | A*02:01 | T |

Variants were not observed in PvSPECT1 from sequences obtained (n = 11)

PbSPECT1 ortholog is conformationally labile and able to change from soluble to membrane-associated form [37,63]. Unlike CelTOS, another sporozoite cell traversal pore-forming microneme protein that is also expressed in ookinetes [64,65], PbSPECT1 is restricted to the liver-stage, specifically salivary gland sporozoites [35–37,63]. Previous work on the cysteine-rich M2 ligand domain of *P. yoelii* MAEBL, using a similar prokaryotic expression system, produced insoluble proteins [34]. Full-length MAEBL is a type-1 transmembrane protein with sequence similarities in its tandem putative ligand domains to the PAN-like ligand domain of apical membrane antigen-1 (AMA1) in the N-terminus and also has similarity to erythrocyte binding protein (*ebl*) family in the C-terminal region [56,57,66–68]. More importantly, MAEBL is expressed in the micronemes of sporozoites and plays an important role in both salivary gland and hepatocyte invasion [31–34,69–71].

In light of these antigens' essentiality to *Plasmodium spp.* liver stage we generated antisera in mice for characterization. High titers were observed against all the recombinant antigens (S2 Fig), although reactivity to whole sporozoites lysate was much lower, except for PvSPECT1 (Fig 3a). The basis of the high seroprevalence to PvSSP3 was unexpected as it is a minor surface protein relative to the immunodominant CSP. However, despite the possible immune distraction by the immunodominant antigen CSP, *in vitro* studies with *P. yoelii* revealed when PyCSP was shed PySSP3 was unmasked [27]. Therefore, we hypothesize that the high seropositive rate we observed indicates that the PvSSP3 could be activated in a similar manner leading to its

high immunogenicity. Indirect immunofluorescence stained sporozoites revealed a surface association of PvSSP3 and PvSPECT1 (Fig 3b and 3c). PvM2-MAEBL was associated with the apical end of the *P. vivax* sporozoites (Fig 3d), similar to other studies. [27,28,30,32,35,36,57] Consistent with this IFA localization, immunoelectron microscopy demonstrated that PbSPECT1 is associated with sporozoite micronemes [35]. Likewise, previous studies have demonstrated MAEBL to be closely associated with sporozoite micronemes in both *P. berghei* and *P. falciparum* [31,32].

To further characterize the immune sera raised against the PE candidate antigens, we evaluated if the anti-PE antisera had inhibitory activity against *P. vivax* sporozoites. Orthologues of these PE antigens have been shown to play important roles in gliding, traversal and invasion [27,28,30,31,33,35]. The ILSDA used primary human hepatocytes and *P. vivax* sporozoites as described previously [49,50]. Each PE-specific antiserum significantly inhibited *P. vivax* hepatocyte invasion (Fig 4a). Interestingly, the mouse antisera to PvM2-MAEBL demonstrated a similar level of inhibition as a previously reported rabbit antiserum to *P. yoelli* M2-MAEBL [30]. These data suggest the presence of functional inhibitory epitopes within the expressed recombinant antigens. Comparative analysis showed an overall decreased growth phenotype for LS forms in all antisera tested (Fig 4b). These data are similar to our previous study that reported a decreased growth phenotype after incubating sporozoites with monoclonal antibodies to CSP repeats of both *P. vivax* and *P. falciparum* as well as antisera to CSP-based *P. vivax* nanoparticle vaccine [50]. These results hint at a post-invasion antibody-mediated mechanism not yet understood.

We next asked the question if our recombinant *P. vivax* PE antigens are naturally immunogenic in *P. vivax* infections by testing post-infection plasma from residents living in a vivax-endemic region of Thailand. PvM2-MAEBL had the lowest overall seropositivity, much lower than PvSSP3 and PvSPECT1, but was significantly different from healthy controls (Fig 5). This was surprising since MAEBL was highly immunogenic in controlled human experimental infection by *P. falciparum* sporozoite inoculation [72]. PvSPECT1 had the highest seroprevalence at ~32% followed by PvSSP3 at 28.8% seropositivity (Fig 5). The high level of seropositivity to PvSPECT1 may be related due to its relative abundance, since it has been identified as one of the most abundant proteins detected in *P. vivax* and *P. falciparum* sporozoites [73–75].

Surface-exposed B cell epitopes are often highly polymorphic as they can be accessible to neutralizing immune antibodies [76,77]. Moreover, such polymorphisms occur as escape mutants are driven by immune selection and the basis of strain-specific immune responses, which is better characterized for *P. vivax* blood-stage antigens [78–81] compared to most *P. vivax* PE antigens. Similarly, variation can be driven in T cell epitopes by abrogating anamnestic responses from prior infections. Additionally, recent systems serology studies highlighted the crucial role of Fc-effector mechanisms in controlling malaria infections [82,83]. Since humoral effector mechanisms can be bolstered by cell-mediated responses as both play a critical role in protection [84–87] we next evaluated the prevalence of variant epitopes that would be indicative of immune selection pressure. FEL was used to detect a diverse selection in a small sample set. Additionally, we also wanted to determine if there is pervasive positive selection by using MEME [51,52]. Positive selection was not detected in our study, probably due to the small sample size. However, a single mutation (D236N), was observed for PvSPECT1 in the MalariaGEN *P. vivax* Genome Variation project's genome browser [88]. We also, identified putative immunogenic epitopes that were consistent with immune selection, using the Immune Epitope Database. For PvM2-MAEBL, a single polymorphic mutation (I981S) was observed in putative B and class I epitopes. Previously for PfMAEBL, the epitope, 'YVSSFIRPDYETKCPPRYPL', was validated to have high affinity to HLA-DRB1*0301 and *1101 and was mostly conserved [89,90]. We identified a similar epitope 'YEEKCPPRFPL' that had an

intermediate binding affinity with an $IC_{50}$ of 101nm & 110 nm for Class I & II, respectively (S4 Dataset).

In this study, we evaluated *P. vivax* PE antigens for an antibody-based vaccine. We provide preliminary characterization of three potential candidates (SSP3, SPECT1, M2-MAEBL). We successfully demonstrated that these *P. vivax* PE antigens expressed as subunit proteins in bacteria are immunogenic in laboratory animals, eliciting functionally inhibitory antibodies that inhibited *P. vivax* liver-stage invasion, and the PE proteins are naturally immunogenic. Altogether our data indicate that these conserved antigens represent suitable targets for *P. vivax* vaccine development. Ultimately, the further evaluation of immunogenicity and protective efficacy of each of these antigens alone or in combination could aid in the development of an effective vaccine against *P. vivax*.

## Supporting information

**S1 Fig. Confirmation of cryopreserved *P. vivax* sporozoite species by ELISA using commercially available monoclonal antibodies.** Raw OD values at 650 nm of sporozoite lysate against mAbs (1 μg/ml) specific to the repeat region of CSP for *P. vivax* VK210, *P. vivax* VK247, *P. falciparum* and *P. berghei*. Error bars indicate s.e.m.
(TIF)

**S2 Fig. Endpoint titers of terminal bleed BALB/c antisera immunized with recombinant antigens.**
(TIF)

**S3 Fig. Distribution of Day 6 *P. vivax* liver-stage small forms ($<$100 μm$^2$) to developing large forms ($>$100 μm$^2$) in the total population.**
(TIF)

**S1 Table. Characteristics of *P. vivax* patients and healthy subjects recruited in this study.**
(DOCX)

**S1 Dataset. Recombinant amino acid sequence of each PE antigen.**
(DOCX)

**S2 Dataset. Mutations identified by variant calling within each of the PE antigens.**
(XLSX)

**S3 Dataset. Codon-by-codon dn/ds of PE antigens.**
(XLSX)

**S4 Dataset. Predicted B and T cell epitopes of PE antigens.**
(XLSX)

## Acknowledgments

We thank the animal facility at the University of South Florida for maintaining the animals. We would also like to thank Naresh Singh and Alison Roth for their contributions in cryopreserving *Plasmodium vivax*, Uriel Ramirez for his help with antigen expression, and Shulin Xu for his assistance in designing the expression constructs of PvSSP3 and PvSPELD.

## Author Contributions

**Conceptualization:** Justin Nicholas, Sai Lata De, Awtum M. Brashear, Patchanee Chootong, John H. Adams.

**Data curation:** Awtum M. Brashear, Samantha J. Barnes.

**Formal analysis:** Justin Nicholas, Samantha J. Barnes.

**Funding acquisition:** Liwang Cui, John H. Adams.

**Investigation:** Justin Nicholas, Sai Lata De, Pongsakorn Thawornpan, Awtum M. Brashear, Surendra Kumar Kolli, Pradeep Annamalai Subramani.

**Project administration:** Samantha J. Barnes, John H. Adams.

**Supervision:** Liwang Cui, Patchanee Chootong, Francis Babila Ntumngia, John H. Adams.

**Writing – original draft:** Justin Nicholas.

**Writing – review & editing:** Justin Nicholas, Sai Lata De, Awtum M. Brashear, Surendra Kumar Kolli, Pradeep Annamalai Subramani, Samantha J. Barnes, Liwang Cui, Patchanee Chootong, Francis Babila Ntumngia, John H. Adams.

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
