## [Decision Letter · Decision Letter 0]

10 Jul 2023

Dear Dr. Adams,

Thank you very much for submitting your manuscript "Preliminary characterization of Plasmodium vivax sporozoite antigens as pre-erythrocytic vaccine candidates" for consideration at PLOS Neglected Tropical Diseases. As with all papers reviewed by the journal, your manuscript was reviewed by members of the editorial board and by several independent reviewers. In light of the reviews (below this email), we would like to invite the resubmission of a significantly-revised version that takes into account the reviewers' comments. 

We cannot make any decision about publication until we have seen the revised manuscript and your response to the reviewers' comments. Your revised manuscript is also likely to be sent to reviewers for further evaluation.

Sincerely,

Gregory Deye

Academic Editor

Charles Jaffe

Section Editor

Reviewer's Responses to Questions

**Key Review Criteria Required for Acceptance?**

**Methods**

-Are the objectives of the study clearly articulated with a clear testable hypothesis stated?

-Is the study design appropriate to address the stated objectives?

-Is the population clearly described and appropriate for the hypothesis being tested?

-Is the sample size sufficient to ensure adequate power to address the hypothesis being tested?

-Were correct statistical analysis used to support conclusions?

-Are there concerns about ethical or regulatory requirements being met?

Reviewer #1: Yes for each.

Reviewer #2: Some statistical analysis missing, otherwise methods are complete and clearly written.

**Results**

-Does the analysis presented match the analysis plan?

-Are the results clearly and completely presented?

-Are the figures (Tables, Images) of sufficient quality for clarity?

Reviewer #1: Yes for each -- minor comments below.

Reviewer #2: (No Response)

**Conclusions**

-Are the conclusions supported by the data presented?

-Are the limitations of analysis clearly described?

-Do the authors discuss how these data can be helpful to advance our understanding of the topic under study?

-Is public health relevance addressed?

Reviewer #1: Yes for each.

Reviewer #2: (No Response)

**Editorial and Data Presentation Modifications?**

Reviewer #1: 1. In the discussion, it is stated “Since humoral effector mechanisms can be bolstered by cell-mediated responses as both play a critical role in protection…” Systems serology studies are indicating the important contribution of fc-mediated antibody functions, whereby the antibodies are providing critical support for cellular responses, for example. This could be just as important or even more important than the more classical “sporozoite neutralizing” functionality used as the over-arching objective for this study. This work should be mentioned in the discussion.

2. When discussing proteins, it would be helpful to always place a “Pv” or “Pb” before the name or mention the parasite species in the sentence or paragraph, so the reader knows what it being referred to. For example, in the Methods section “Antigen Production”, the manuscript doesn’t specify what parasite this is, although of course Pv is implied. In the corresponding Results section, it does say “P. vivax” in the heading. In line 412 it does not. This would just help smooth the reading experience.

3. Line 88: R21/Matrix-M is now licensed in two African countries, Ghana and Nigeria.

4. Line 104: suggest “in hepatocytes” rather than “in the liver”

5. Table 1, last entry – To my understanding, the Al-Nihmi reference showed that a parasite that arrests during liver stage immunity (based on knock out of SPELD) shows that (a) SPELD is needed for liver stage development and (b) sporozoites that arrest during liver stage development can induce protective immunity, which is of course not a new finding. I don’t think it shows that SPELD itself is protective. Perhaps this should be reworded as something like “pbspeld ko sporozoites arrest development during the early liver stages” emphasizing the importance of the antigen rather the fact that sporozoites are a protective immunogen which is irrelevant to the manuscript.

6. Figure 3a: I’m thinking it would be better to plot all the values for non-immunized mouse sera as individual data points in their own cloud rather than drawing the dotted line. It doesn’t look as if there is any difference between MAEBL-immunized mice and non-immunized mice. Also suggest that you use the technique of horizontal brackets at the top indicating the p-value for various comparison among the (it would then be four) clusters of data points (this technique is used in Figure 4). Also, if you are wishing to define seroconversion (dotted line), there should be some definition such as x times the median of controls or x standard deviations above controls or something similar specified in the figure legend.

7. Line 455: there is an extraneous “if” that can be omitted.

8. The manuscript mentions in several places that one approach to developing an effective vaccine would be to combine antigens. Were the antisera from these antigens looked at in combination?

Reviewer #2: (No Response)

**Summary and General Comments**

Reviewer #1: This is well written manuscript characterizing four potential Pv vaccine candidates that would function by inducing antibodies inhibiting parasite migration, hepatocyte invasion and liver stage development during the pre-erythrocytic stages of infection. The methods and results are clearly described and presented, and I have no major criticisms. The discussion and conclusion are appropriate. I feel that this work is a significant contribution to the literature and is especially important due to the dearth of effort on Pv, which, ultimately, may turn out to be the most difficult malaria parasite to control (excepting zoonotic Pk).

Reviewer #2: The authors present a concise and interesting manuscript detailing the preliminary evaluation of several P. vivax pre-erythrocytic candidate antibody targets. The need for this work is high and therefore the results are of high relevance both to identify targets and identify the best means of vetting these targets preclinically given the paucity of tools for Pv. Unfortunately, serious concerns regarding the functional data keep this manuscript from being of sufficient quality to be of impact for the community. This includes unclear presentation and analysis of data as well as opaque interpretation. Critical controls are also not included. Combined with unclear presentation of some biological aspects of invasion and infection the data are overall very difficult to use to extract actionable information.

Major Critiques

1. Thank you for the methods section. This will be an asset to the community.

2. Can the authors please show more than a single sporozoite for the IFAs, please? This would make their conclusions about protein localization stronger.

3. There are serious issues with the ILSDA data. 

a. First, how inhibition is quantified is not described clearly. Second, description of each readout is obtuse with “inhibition”, “reduced liver stage burden”, “liver stage growth and development was significantly attenuated”, “average schizont area”, and “growth attenuation” all used somewhat interchangeably and without specific references to which measure (e.g. reduction in overall liver stages, reduction in schizonts only, reduction in schizont area) is being utilized.

b. Furthermore, looking at Table S2, it is unclear how any reductions are seen either in schizonts or overall burden. All conditions have numbers of large forms around or greater than the average of the “no pAb” control of 114. 

c. The figures seem to make comparisons between test antigen groups and DBPII for Fig 4a which is appropriate while Fig 4b compares to so serum control. The text references differences between no serum control for significance only. However, the text also notes a large impact of the DBPII serum (which should be a control for liver stages) on growth which suggest the results cannot be attributed to anything but the effects of adding mouse serum. Can the authors please clarify these inconsistencies?

d. There also appears to be no positive control using an anti-PvCSP210 mAb as previously published by this group. Why is this? 

e. Finally, can the authors expand on why there is no impact on hypnozoites/small forms? This is intriguiging and in contrast to published work with anti-CSP mAbs.

4. ELISA titers from the immunized mice should be shown even if in supplemental.

5. If both MAEBL and SPECT1 are micronemal, why do they have different staining patterns? Micronemal localization is used to justify both the surface staining of SPECT1 and apical localization of MAEBL. 

Minor Critiques

6. Line 94: This sentence is difficult to understand. Is the suggestion that PvCSP in a multivalent (ie with other antigens) would increase immunogenicity? In addition, “poor immunogenicity” cannot be caused by a lack of long-lived responses rather they a lack of long-lived responses are a result/indicator of poor immunogenicity.

7. In Line 112-114, the authors state that “gliding is essential for infectivity” yet cite a publication (REF 25) in which SSP3- sporozoites are gliding deficient yet perfectly capable of infection in vivo. The next references describe parasites with varying levels of gliding with inconsistent effects on infectivity. If there is a more specific scenario in which gliding is essential, could the authors please specify such that the biological importance of in vitro gliding can be better interpreted by the uninitiated reader?

8. Similarly, the terms “gliding”, “invasion” and “infection” are used ambiguously. These are nuanced parts of the complicated parasite life cycle and therefore should be used with more precision and consistency.

9. Table I, column 1, “essentiality to invasion” is a bit misleading and would be better served by “putative function” or similar. Likewise “Association with infectivity” appears to rather be “evidence as protective immune target” 

10. This is a nitpicky nuance, but since the sporozoites being used are fixed prior to staining, this is not exactly “native”. 

11. Why were different t-tests used in Fig. 5? Related to this, based on the binary classification of responders vs. non-responders (using RI of 1) and the Results section which describes results in terms of the percentage responding, a proportional statistical test is likely more appropriate rather than a t-test which would be appropriate for a titer-based comparison.

12. The last line (507-509) should read something like: “the further evaluation of the immunogenicity…could aid in the development”. Given that the functionality data of this manuscript are thin and do not include any combinatorial effects this should remain a speculative statement.

PLOS authors have the option to publish the peer review history of their article (what does this mean?). If published, this will include your full peer review and any attached files.

Reviewer #1: No

Reviewer #2: Yes: Brandon Wilder
---

## [Editor Report · Decision Letter 1]

15 Aug 2023

Dear Dr. Adams,

We are pleased to inform you that your manuscript 'Preliminary characterization of Plasmodium vivax sporozoite antigens as pre-erythrocytic vaccine candidates' has been provisionally accepted for publication in PLOS Neglected Tropical Diseases.

Best regards,

Gregory Deye

Academic Editor

Charles Jaffe

Section Editor

---

## [Editor Report · Acceptance letter]

8 Sep 2023

Dear Dr. Adams,

We are delighted to inform you that your manuscript, "Preliminary characterization of Plasmodium vivax sporozoite antigens as pre-erythrocytic vaccine candidates," has been formally accepted for publication in PLOS Neglected Tropical Diseases.

Best regards,

Shaden Kamhawi

co-Editor-in-Chief

Paul Brindley

co-Editor-in-Chief
